# High Frequency of *Apodemus* Mice Boosts Inverse Activity Pattern of Bank Voles, *Clethrionomys glareolus*, through Non-Aggressive Intraguild Competition

**DOI:** 10.3390/ani13060981

**Published:** 2023-03-08

**Authors:** Remo Probst, Renate Probst

**Affiliations:** Ornis—Biology Engineering Office and Research Institute, Dr. G. H. Neckheimstr. 18/3, A-9560 Feldkirchen, Austria

**Keywords:** *Apodemus* spp., *Clethrionomys glareolus*, activity pattern, camera traps, intraguild competition, interspecific aggression, temporal partitioning, niche separation, mast year, Alps

## Abstract

**Simple Summary:**

Ecologically similar animals can only co-occur if resources are used in a sufficiently diverse manner. The necessary separation can be made according to habitat, food choice and activity pattern. We investigated this question in an inner-alpine mixed forest for bank voles, *Clethrionomys glareolus*, and mice of the genus *Apodemus*, the latter being larger and stronger. In the presence of many *Apodemus* mice, bank voles were indeed more diurnal. We found no alternative explanation for this temporal partitioning, despite considering possible other influencing factors, such as exposure to lunar illumination, temperature, precipitation, the presence of predators or the availability of cover. Our video footage showed that this temporal separation hardly requires any physical interaction between the small forest rodents. We attribute this to the long common evolution of these animals.

**Abstract:**

Sympatric animals with similar requirements can separate their ecological niches along the microhabitat, food and time axes. There may be alternative reasons for an interspecific different activity pattern, such as intraspecific social constraints, predator avoidance or physical conditions such as temperature, precipitation and illumination. We investigated the importance of intraguild competition in a 2-year study in an inner-alpine mixed forest, using small forest rodents as our model species. *Apodemus* mice were the physically superior, and bank voles, *Clethrionomys glareolus*, the inferior competitor. We predicted that bank voles would exhibit increased diurnal activity when frequencies of the almost exclusively nocturnal *Apodemus* mice were high during a seed mast year. To investigate this, we recorded 19,138 1 min videos. Controlling for confounding variables, bank vole diurnal activity was significantly related to the frequency of *Apodemus* mice. We assume that at high densities of *Apodemus* mice, a purely nocturnal separation in the niche dimensions of time, habitat and microhabitat is no longer sufficient, and therefore an inverse activity pattern by the bank voles is reinforced. Our videos showed, however, that this does not require persistent aggressive meetings and we explain this by the long co-evolution of the taxa under study.

## 1. Introduction

Animals are exposed to a plethora of environmental parameters in their habitat. For small mammals, such as bank voles (*Clethrionomys glareolus*, formerly *Myodes glareolus* [1]) and *Apodemus* spp. mice (further referred to as *Apodemus* mice), these are primarily physical conditions such as temperature, precipitation and illumination [2,3,4,5], as well as biotic factors such as food, predation, intraspecific organisation and interspecific competition [6,7,8,9,10,11,12].

Competition among species is a widespread phenomenon [13,14], and long-term coexistence is only possible if at least one important niche dimension differs. Dimensions of resource partitioning are mainly habitat, food and temporal niche [15,16]. For bank voles and *Apodemus* mice, there are indications for competitive mitigation on all three niche axes, microhabitat [9,17,18,19], nutrition [7] and chronoecology [4,20,21]. The stronger and larger *Apodemus* mice are considered superior to bank voles [20,22].

Activity time has long been known as a niche axis [16,23], however, the relevance of the temporal niche concept has been much debated [15,24]. In several studies, only weak interspecific competition or the resulting time separation between *Apodemus* mice and bank voles was found [25,26,27], although *Clethrionomys glareolus*, in contrast to the nocturnal *Apodemus* mice, are frequently diurnal [28,29]. Main problems addressed are the difficulty of directly observing competition and the need to take into account a wealth of confounding factors potentially influencing activity time [13,25]. To overcome these methodological problems, we conducted a 2-year camera-trapping study in the southern part of the Alps, capturing the forest-dwelling *Apodemus* mice and bank voles with the available video function. Camera-trapping is a powerful tool for wildlife research [30,31,32,33,34], including behavioural investigations of difficult-to-observe species [35,36]. Thus, the diurnal activity of bank voles could be tested against predictor variables, such as the activity of *Apodemus* mice and predators, habitat characteristics and abiotic parameters. Further, the type of interactions between the two small forest rodent taxa, ranging from tolerant meetings to aggressive displacements [20], could be studied.

An important prerequisite for comparing the activity patterns of bank voles and *Apodemus* mice is the variation in frequency. Small forest rodents show the greatest population changes in the context of masting, especially among temperate tree species. Masting is the pulsed, synchronous production of large seed crops, which can dramatically affect the consumer community [37,38,39]. Populations of *Apodemus* mice show a pronounced outbreak–crash pattern in response to the seed masts [10]. The findings are less clear for the more generalist bank vole [7,40,41,42], where cyclic, stable and intermediate population dynamics have been found across Europe [40]. The first year of our study was characterised by an extreme number of pollen not yet observed in the 30-year recording history of the Austrian federal state of Carinthia [43]. This was especially true for locally dominating woody plants, such as the Norway spruce, *Picea abies*, European beech, *Fagus sylvatica*, and European hazelnut, *Corylus avellana*. The positive connection between airborne pollen and quantity, as well as quality of seed crop in these so-called masting pollen producers could be proven elsewhere [44,45,46,47]. The second year of the investigation was characterized by a nil crop. In accordance with these particular differences in the availability of plant seeds, we will hereafter refer to the first year as the mast year and the second as the non-mast year.

In most of central European forest ecosystems, bank voles and *Apodemus* mice are of outstanding importance and may be designated as keystone species. They are an essential food for a number of predators [8,48,49], can themselves act as seed consumers, as well as dispersers in context-dependent plant-granivore interactions [50,51,52], may damage forest trees [53,54], and can be vectors of a number of diseases [55,56].

In our study we expected a sharp increase in small rodent numbers after the mast pulse and a subsequent collapse of populations, which would particularly affect the specialised seed-eating *Apodemus* mice. The variations in frequency would be a prerequisite for behavioural changes (Assumption 1). As a second important prerequisite for the shift of the bank voles into diurnal activity due to intraguild competition, we expected an almost exclusive nocturnal activity of *Apodemus* mice (Assumption 2). Controlling for confounding abiotic and biotic parameters, temporal partitioning would be an indirect measure for reducing intraguild competition. We predicted asymmetric relationships, whereby bank voles would show an inverse activity pattern with an increased presence of *Apodemus* mice (Prediction 1). As a direct measure, we predicted an altered antagonistic behaviour, with higher numbers of *Apodemus* mice leading to increased interspecific aggressions against bank voles (Prediction 2).

## 2. Materials and Methods

### 2.1. Species Identification

While bank voles are fairly easy to identify visually on appropriate videos, for example by the ratio of body to tail length or the size of the ears [29], we have not attempted to distinguish members of the genus *Apodemus*. In the latter, identification problems can occur without calibration to the morphology of the local population at hand [57], and even genetic identification has proven difficult [58,59,60]. As *Apodemus* mice were typically silent, identification by means of the calls was also ruled out [61]. Due to the well-known habitat requirements [28,62], we expected the occurrence of the yellow-necked mouse, *Apodemus flavicollis*, and of the wood mouse, *Apodemus sylvaticus*, as most probable in our forested study area, in an unknown numerical ratio. The alpine mouse, *Apodemus alpicola*, is absent from the region according to current knowledge [63].

### 2.2. Study Area

The camera traps were set up on the mountain range of the Ossiacher Tauern (46,692° N; 14,067° E, 550 m a. s. l.) in Carinthia, Southern Austria. In the two years of the survey, the mean annual temperature was 8.88 °C and 9.39 °C, respectively, and the total annual precipitation was 1159 mm and 749 mm, respectively (meteorological station of Feldkirchen: https://data.hub.zamg.ac.at/ (accessed on 15 October 2022)). The study area was on the steep lower slope with a distinctly undulating relief and several small rocks. The concave areas have moist and humus-rich soils (sandy loams over clay of the pseudogley type), where a lime-maple noble deciduous forest (Tilio-Acerion) would be the primary vegetation. In the convex parts, beech forest types would predominate plant-sociologically (Melampyro-Fagetum, Galio odorati-Fagetum [64]). European beech, *Fagus sylvatica*, as the actual primary tree species has been pushed back, and today Norway spruce, *Picea abies*, dominates the forest. However, European beech, limes, *Tilia platyphyllos* and *T. cordata*, sycamore, *Acer pseudoplatanus*, European ash, *Fraxinus excelsior*, as well as silver fir, *Abies alba*, are mixed in, and therefore the study site is classified as a mixed deciduous and coniferous forest biotope type. Trees with a low trunk diameter dominate, but there are individual “achievers”, as well as sections of pole wood and pioneer stages, the latter especially consisting of the European hazelnut, *Corylus avellana*. The study site is not located in any state-protected area.

### 2.3. Camera Trap Study

#### 2.3.1. Study Period

Camera traps were set up from September 2020 to September 2022.

#### 2.3.2. Type and Model

Wild-Vision Full HD 5.0 with Black-LED flash was utilized for the study. The trigger speed was lower than 1 s. The passive infrared sensor (PIR) was designed for high sensitivity.

#### 2.3.3. Modifications

To reduce the over-illumination that occurs when animals are close to the camera trap, half of the LED field was switched off. We changed to 64 MB cards in order to store the large amount of video data.

#### 2.3.4. Recordings

We used 1 min videos in HD resolution of 1280 × 720 pixels. Only those recordings where there was no equipment failure, user error, or false trigger were used for further analysis.

#### 2.3.5. Definition of an Event

All our specifications refer to the recorded 1 min videos. Several individuals of one species are only counted if they were visible on a single video and at the same time.

#### 2.3.6. Bait

*Apodemus* mice and bank voles were attracted with unpeeled, black-and-white sunflower seeds. Each deployment equated to 0.5 kg per camera trap. The weight of the sunflower seeds presented was based on experimental results within the genus *Apodemus*. At high densities, several hundred grams can be consumed and transported from a single feeding site, especially during the intensively used first night [65]. In order to address the questions of relative frequency and activity pattern, it had to be possible to feed a maximum number of target species individuals for a sufficient period of time. The sunflower seeds were provided on the ground and not covered. This enabled both a quick triggering of the camera traps and the most reliable identification of small mammals, as well as other species.

#### 2.3.7. Layout

According to a minimum convex polygon analysis of camera trap sites by means of QGIS 3.18.0, the study area had a size of 4.8 ha. A random allocation was carried out [66], with 83 sites being sampled over the course of the investigation.

#### 2.3.8. Spatiotemporal Independence

In order to get a high degree of independence between trials, we set up camera traps every 12.07 ± 6.32 sd days [67], and sample sites from the last trial were not allowed to be repeated. Further, the analyses were standardised according to the first 24 h of video recording per sampling site [68], hereinafter referred to as a camera trap-night. To minimise the movement between two camera traps and the immigration of small mammals within one trial, a distance of 58.94 ± 42.47 sd m was maintained between typically 6 simultaneously active cameras. Distances were determined with QGIS 3.18.0 and corrected for the slope.

#### 2.3.9. Camera Trap Placement

The bait was centred in the middle of the shot and sunflower seeds were spread over a maximum area of 50 × 50 cm. This distance from the focal point to the camera (1–2 m) and the height of the camera (typically from 0.25–0.75 m), and thus the angle, depended on the different slopes in our study area. Camera traps were only placed in locations where minimal vegetation change was required for a clear line of sight to the bait.

### 2.4. Data Treatment

For the analysis of day–night activity pattern, we achieved a high degree of independence of the data by keeping only the single recording with the highest number of *Apodemus* mice or bank voles from all videos of the same camera trap within a 30 min period [69]. The 48 periods of 30 min periods within a calendar day were assigned to either day or night, separated by sunrise or sunset. The two twilight periods of a day were assigned to the time of day in which more of the 30 min were located.

### 2.5. Predictors

For the multivariate regression analysis of the effects of extrinsic factors, we tested 6 additional independent variables besides mast year/non-mast year: (1) lunar illumination: % illumination in the camera trap-night (https://moonphases.co.uk/ (accessed on 10 December 2022)); (2) minimum temperature: average of the minimum values of the two calendar days of one camera trap-night (https://data.hub.zamg.ac.at/ (accessed on 12 December 2022)); (3) precipitation at night: precipitation total (mm) from 19:01–7:00 CET per camera trap-night (https://data.hub.zamg.ac.at/ (accessed on 12 December 2022)); (4) predators: number of 30 min periods per camera trap-night with at least one visible predator on the available videos. In our dataset, red fox, *Vulpes vulpes*, stone marten, *Martes foina*, European polecat, *Mustela putorius*, and tawny owl, *Strix aluco*, were classified as mandatory predators of small mammals. Moreover, (5) cover: for our year-round comparison, we recorded the % of cover with lying deadwood, snags and rocks in a 10 m radius of the camera trap, because numerous studies have shown a positive correlation between the abundance of voles of the genus *Clethrionomys* and the amount of woody debris and brushwood coverage (so-called thigmotaxis [70,71,72]); and (6) *Apodemus* spp.: Σ of *Apodemus* mice from the pseudo-replication-limited 30 min periods per camera trap-night.

### 2.6. Statistical Analysis

The data were statistically analysed by the program package R 4.2.2 (https://www.R-project.org/ (accessed on 14 December 2022)). Mann–Withney U tests were used to examine the frequency differences between two groups, and χ^2^ tests to examine the relationship between nominal variables. The dichotomous day–night activity of bank voles was investigated by multivariate logistic regression. Activity patterns were analysed with Hermans–Rasson tests, using the R-package “CircMLE” [73,74]. Overlap coefficient (Δ) between the temporal activity patterns of *Apodemus* mice and bank voles was estimated with the R-package “overlap” [75]. This coefficient ranges between 0 (no overlap) and 1 (complete overlap, [76]). Since the number of cases for the comparisons was greater than 75, the Δ_4_ estimator was calculated [76]. In addition, 95% confidence intervals for Δ_4_ using 10,000 bootstraps were determined. The overlap was rated as “medium” when Δ_4_ was between 0.50 and 0.75, as “high” with Δ_4_ > 0.75 and as “very high” if Δ_4_ was larger than 0.90 [77]. Statistical significance was set at *p* < 0.05, and marginal significance at *p* < 0.10.

## 3. Results

### 3.1. Frequency of Small Forest Rodents

On 215 camera trap-nights, 19,138 analysable videos (319.0 h) were taken within the two years of survey. 102 camera trap-nights were conducted in the first mast year, and 113 in the non-mast year. At least one small forest rodent was detected on 15,931 1 min videos (265.5 h). Using the maximum number of either simultaneously visible bank voles or *Apodemus* mice per camera trap-night as a measure of frequency, population trends between the mast year and the non-mast year could be compared (Figure 1). The frequency of bank voles did not differ between the mast year and the non-mast year (U = 5965, *p* = 0.63, r_g_ = 0.035). Bank voles showed a tendency to be more frequent after the reproductive phase from spring into summer in both survey years, but three, or in a single case four, individuals visible on the same video always represented an exception. In contrast, *Apodemus* mice were significantly more frequent in the mast year compared to the non-mast year (U = 9297, *p* < 0.001, r_g_ = 0.612). In the mast year, *Apodemus* mice overwintered in high numbers and reached their population peak the following summer. Regularly up to five, and exceptionally up to eight, individuals could be counted on the videos at the same time. In the non-mast year, the population collapsed totally and recovered only slowly towards the end of the study, in the second summer. In winter and spring of the non-mast year, *Apodemus* mice were not detected on many camera trap-nights and were rarer than bank voles during this phase (Figure 1).

### 3.2. Activity Patterns of Small Forest Rodents

Activity patterns were analysed using maximum rodent numbers per 30 min intervals. According to Hermans–Rasson tests, activity patterns were significantly different from random in the bank voles (t = 25.58, *p* = 0.01) and *Apodemus* mice (t = 161.49, *p* = 0.01). This was also valid when the mast year (bank voles: t = 25.79, *p* < 0.001, *Apodemus* mice: t = 1159.60, *p* < 0.001) and the non-mast year (bank voles: t = 30.98, *p* < 0.001, *Apodemus* mice: t = 464.90, *p* < 0.001) were tested separately. Bank voles showed a predominance of diurnal activity in the mast year (56.5%), while nocturnal activity predominated in the non-mast year (62.6%). The association between diurnal activity and mast year was significant (χ^2^ (1) = 47.029, *p* < 0.001, Cramérs V = 0.188). *Apodemus* mice were overall much less diurnal than bank voles (mast year: 6.3%, non-mast year: 2.7%), but again, diurnal activity was significantly higher in the mast year (χ^2^ (1) = 13.14, *p* < 0.001, Cramérs V = 0.072). Over both study years, the “medium” overlap of activity rhythms between bank voles and *Apodemus* mice was 59% (Δ_4_ = 0.591, 95% CIs = 0.564–0.619). The overlap was noticeably lower in the mast year (Δ_4_ = 0.501, 95% CIs = 0.463–0.538) than in the non-mast year (Δ_4_ = 0.647, 95% CIs = 0.609–0.684).

### 3.3. Temporal Niche of Bank Voles

A multivariate logistic regression was used to investigate the dichotomous activity pattern of *Clethrionomys glareolus* against seven extrinsic predictors (Table 1). The probability of nocturnal activity was significantly increased in the non-mast year (*p* < 0.001, OR = 2.329). Overall precipitation during the night significantly reduced activity (*p* < 0.001, OR = 0.932). A marginal significant effect was caused by lunar illumination (*p* = 0.051, OR = 0.997) and the activity of *Apodemus* mice (*p* = 0.060, OR = 0.992), both of which led to a tendency towards increased daytime activity.

In the mast year, univariate comparisons revealed significantly more diurnal activity of bank voles, with an increased number of *Apodemus* mice (U = 41,794, *p* = 0.01, r_g_ = 0.125), whereas this was not the case in the non-mast year, with a much lower absolute number of interspecific competitors (U = 76,515, *p* = 0.09, r_g_ = 0.072). Nocturnal precipitation induced a significant increase in diurnal activity of bank voles in both the mast year (U = 39,366, *p* = 0.045, r_g_ = 0.059) and the non-mast year (U = 78,993, *p* < 0.001, r_g_ = 0.106). Increased lunar illumination resulted in a significant increase in nocturnal activity in the mast year (U = 32,337, *p* = 0.008, r_g_ = −0.130), but in a significant decrease in nocturnal records of bank voles in the non-mast year (U = 86,762, *p* < 0.001, r_g_ = 0.215).

### 3.4. Intraguild Aggression

*Apodemus* mice and bank voles were only seen together on 350 videos out of 15,931 recordings, with at least one of these small forest rodents filmed (2.20%). Encounters were about equally frequent in the non-mast year and in the mast year, although in the latter *Apodemus* mice (but not bank voles) were much more frequent (Table 2, comp. Figure 1). In the mast year, *Apodemus* mice were less aggressive, whereas in the non-mast year bank voles more often avoided *Apodemus* mice or were actively chased away from the food by them (χ^2^ (2) = 7.240, *p* = 0.027, Cramérs V = 0.144).

## 4. Discussion

Rodent communities are considerably shaped by interspecific competition for limited resources and temporal partitioning is one possible dimension of niche separation [16,78,79]. We studied whether bank voles show an inverse activity pattern compared to the superior *Apodemus* mice. There are important, but contradictory, studies on this question [4,20,21,25,80,81,82,83,84,85,86,87,88,89], and there is still no consensus on the influence of interference competition on the chronoecology of the bank voles due to the use of different methods (e.g., capture data, cafeteria experiments, and camera-trapping), the incomplete inclusion of confounding variables, the lack of year-round studies and the limited knowledge on the frequency, as well as nature of direct confrontations with *Apodemus* mice. We tried to approach this issue with a large sample size of videos from camera traps in an inner-alpine mixed forest, which were baited to increase the number of observations [20]. We assume that our smell or handling hardly delayed the use of the bait, because (diurnal) bank voles regularly appeared in the first half hour after setting up the camera trap (19.2 ± 7.66 sd min, minimum 3 min, *n* = 10).

Assumption 1: An important requirement to test competition among species is met if at least one species is particularly common [9]. We expected the massive increase of small forest rodents in the seed mast year. In agreement with many other studies [10], this first assumption was fulfilled in *Apodemus* mice. In contrast, bank voles remained at low population levels throughout the study. Such relatively stable populations have been found, for example, in the transition zone between the northern cyclic and the southern outbreak populations in the so-called “limes norrlandicus” in Scandinavia [40], in undergrowth-poor forest stands of the central European lowlands [42] and on low-productivity study areas of the inner Alps [90]. For our research question, this ideally meant a quasi-experimental variation in the frequency of the dominant competitor, with the frequency of the inferior species remaining approximately constant. The small number of bank voles also renders (largely unknown) intraspecific effects unlikely to outweigh interspecific competition [91].

Assumption 2: Albeit *Apodemus* mice were more diurnal in the mast year, they were almost exclusively nocturnal overall. This is in agreement with numerous findings on activity patterns within the genus *Apodemus* [4,21], which we could not distinguish at a species level in our study. Bank voles showed more variability, with diurnal activity predominating in the mast year, but nocturnal activity predominating in the non-mast year. Accordingly, the overlap in activity rhythms between the two small rodent taxa was much smaller in the mast year, when *Apodemus* mice were common. This suggests an avoidance strategy, which, however, remains a vague assumption without the inclusion of other possible explanatory variables.

Prediction 1: We predicted an inverse activity pattern of bank voles with an increased presence of *Apodemus* mice, controlling for a set of other possible explanatory variables. The analyses revealed a close relationship between the frequency of *Apodemus* mice and the diurnal activity of *Clethrionomys glareolus*, particularly and significantly in the mast year. This indirect measure strongly suggests a temporal partitioning effect due to intraguild competition, because none of the numerous additional explanatory variables describe the same relationship: overnight precipitation reduced bank vole nocturnal activity, regardless of the frequency of *Apodemus* mice. This is consistent with findings in other studies [3,82], but contrary evidence exists [2] and causes are not fully understood (e.g., thermoregulatory costs, interspecific concurrence, and predation risk). Lunar illumination led to an overall marginally increased diurnal activity of the bank voles, but interestingly with different effect directions in the mast and the non-mast years. While bank vole night activity increased significantly with lunar illumination in the mast year, it was significantly reduced in the non-mast year. We suggest an association to intraguild competition in this case as well, because *Apodemus* mice are specifically known for their lunar phobia [3,4,19]. Therefore, bank voles might take advantage of the relatively competition-free period of bright lunar nights, as *Apodemus* mice reduce their activity even at high densities due to bright illumination. At low *Apodemus* mice densities, bank voles may use even the darkest hours to avoid visual predators [92,93]. This would explain the contrasting findings on this physical factor in other studies [3,4,19].

For the non-significant predictor cover, we assume too much uniformity between camera trap locations. To the bank voles’ disadvantage [9,18,26,42,94,95,96], our study forest was generally poor in cover below the tree layer, which is probably reflected in the comparatively low frequency of this species, even in the mast year [62,97]. We detected very few predators on our video recordings (*n* = 7). All were nocturnal, almost exclusively mammalian predators (85.7%) and occurred primarily in the non-mast year (87.5%), where a time-delayed population increase of these generalist, non-migratory, and territorial predator species can be expected as a consequence of dispersal. Therefore, it is even unlikely that indirect cues from predators, such urine, faeces and anal gland secretions [8,93,98,99,100,101], led to increased daytime activity in bank voles in the mast year. In addition, there was also no discernible correlation between the minimum temperature and the behaviour of the bank voles. In other studies, the effects of temperature were small as well or counteracting [2,3,82].

Finally, it is particularly important to note that the independent variable mast year/non-mast year itself is significantly correlated with the diurnal activity of bank voles. Since a test for multicollinearity showed that predictors of the multivariate logistic regression model were not redundant (each variance inflation factor < 2), there must be an influence on bank vole diurnal activity pattern beyond the mere frequency of *Apodemus* mice. Having excluded the most common alternative variables, we surmise an additional influence of the other two niche axes: habitat and food. It is well known that the dietary niche between these small forest rodents overlaps much more in mast years, and high densities of *Apodemus* mice make spatial separation more difficult for the bank voles, even at the microhabitat level [41,102,103]. We therefore assume that under such ecological preconditions, the diurnal activity becomes particularly important for *Clethrionomys glareolus*. Due to a similar food choice and the “omnipresent” *Apodemus* mice, raised fitness costs of diurnal activity (e.g., increased predation risk [104,105]) can no longer be compensated for at night.

Prediction 2: The two taxa encountered each other very rarely in front of the camera traps. The prediction that a higher number of *Apodemus* mice would lead to increased intraguild aggression towards bank voles was not confirmed. In the non-mast year, despite increased aggressiveness of *Apodemus* mice, no inverse activity pattern of bank voles analogous to the mast year was observed. Thus, the greater number of *Apodemus* mice had a stronger negative impact on bank voles in the mast year than could be compensated by improved food supply. As the number of encounters did not differ between the mast year, with numerous *Apodemus* mice, and the non-mast year, we assume an active avoidance strategy by the bank voles. Given the long co-evolution of these small forest rodents, it would be unlikely that dominance relationships need to be actively re-established permanently. Other authors, too, could not find evidence for persistent aggressive meetings between *Apodemus* mice and bank voles [20,25,27]. In the non-mast year, the few adult *Apodemus* mice (as far as assessable by body size) were particularly aggressive, probably in the struggle for scarce food resources.

## 5. Conclusions

Our studies showed that a high frequency of the almost exclusively nocturnal *Apodemus* mice led to increased diurnal activity of the bank voles, and this could not be explained by an extensive set of alternative predictors. We assume that at high *Apodemus* mice densities, increased separation along the time axis by bank voles must occur, as at night, there is no longer sufficient relief of intraguild competition in all niche dimensions: time, microhabitat and diet. The “omnipresence” and intimidation by *Apodemus* mice is satisfactory for the bank voles’ inverse activity pattern, without the need for frequent aggressive meetings. For rigorous quantification of the exclusive influence of *Apodemus* mice on the activity budget of bank voles, studies in areas where the two genera do not live syntopically, and extensive, capture-mark-recapture or removal experiments simultaneous to the camera trap survey, would be purposeful.

## Figures and Tables

**Figure 1 animals-13-00981-f001:**
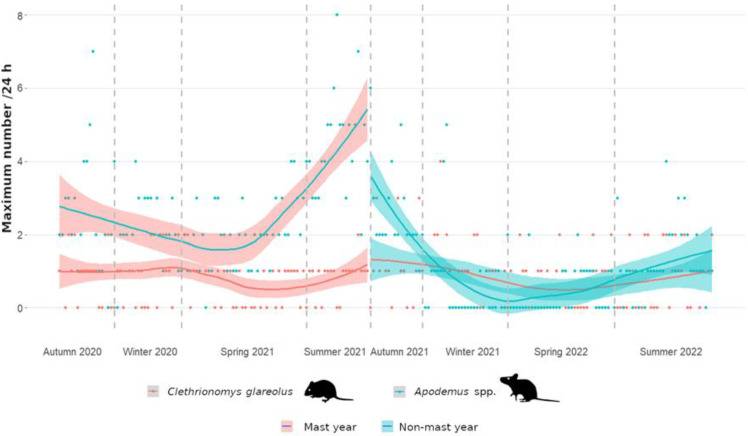
Population development of small forest rodents in an inner-alpine mixed forest. The frequency of *Apodemus* mice increased significantly in the mast year, whereas this was not the case for bank voles, *Clethrionomys glareolus*. In the non-mast year, the population of *Apodemus* mice collapsed.

**Table 1 animals-13-00981-t001:** Effects of seven environmental predictors on the nocturnal activity of bank voles, *Clethrionomys glareolus*.

	Estimate	Standard Error	Z-Score	*p*-Values	Odds Ratio	2.50%	97.50%
(Intercept)	0.186	0.163	1.139	0.255	1.205	0.874	1.660
Non-mast	0.845	0.132	6.426	<0.001	2.329	1.801	3.017
Lunar Illumination	−0.003	0.001	−1.949	0.051	0.997	0.994	1.000
Temperature_Min_	0.002	0.008	0.217	0.828	1.002	0.987	1.017
Precipitation_Night_	−0.07	0.020	−3.551	<0.001	0.932	0.896	0.969
Predators	−0.219	0.273	−0.802	0.423	0.803	0.470	1.374
Cover	−0.061	0.049	−1.243	0.214	0.940	0.853	1.036
*Apodemus* spp.	−0.009	0.005	−1.884	0.060	0.992	0.983	1.000
Nagelkerkes R^2^ = 0.083							

**Table 2 animals-13-00981-t002:** Aggressive behaviour of *Apodemus* mice towards bank voles, *Clethrionomys glareolus*, distinguished between the mast year and the non-mast year.

	No Aggression	Avoidance	Chase	Total
Mast Year	74 (43.0%)	77 (44.8%)	21 (12.2%)	172 (49.1%)
Non-mast Year	52 (29.2%)	99 (55.6%)	27 (15.1%)	178 (50.9%)

## Data Availability

Not applicable.

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
