# Peer review of "High Frequency of *Apodemus* Mice Boosts Inverse Activity Pattern of Bank Voles, *Clethrionomys glareolus*, through Non-Aggressive Intraguild Competition"

_animals, 2023, doi:10.3390/ani13060981_

Round 1

Reviewer 1 Report

The authors of this paper investigated whether intraguild competition between mice belonging to the genus Apodemus and the bank vole Clethrionomys glareolus can lead to increased diurnal activity of bank voles (called inverse activity pattern). The two-year study was conducted in Carinthia, southern Austria, in an inner-alpine mixed forest. The authors assumed that bank voles would show increased diurnal activity when the number of nocturnal Apodemus spp. mice was high in a seed-abundant year. To achieve the objectives of the study, the authors used camera recording and tested the influence of seven external factors on the nocturnal activity of bank voles.

Although this issue has already been studied by several researchers, it still remains incompletely understood, and not all factors that can cause an inverse activity pattern in bank voles have been identified. Therefore, I consider it reasonable to undertake a study of this issue in order to better understand it.

The article is well written and well structured. The introduction identifies the issue well by referring to similar studies conducted to date. The authors set the objectives of the research in the form of 4 predictions, which are then verified (lines 87-98). While the last two research goals actually relate to predicting future events (and therefore can be called predictions 3 and 4), predictions 1 and 2 are rather assumptions/prerequisites that need to be met in order for predictions 3 and 4 to be verified (the answers to predictions 1 and 2 are rather obvious in the light of previous studies published by other authors). Therefore, I would use assumptions or prerequisites instead of predictions 1 and 2 (in other words, to verify predictions 3 and 4, it is required to meet the prerequisites, now called predictions 1 and 2). Regarding statistical analysis, the collected research material was analysed using well-chosen statistical methods. In the following sections of the article, the authors presented and interpreted the results obtained correctly, related their results to those reported by other authors, and drawn the appropriate conclusions. Overall, the article is interesting and well prepared. However, there is one thing that may raise doubts and concerns, and therefore should be clarified - the use of bait (sunflower seeds) to lure studied species to the sites monitored by the video cameras. As the authors explain (line 295), bait was used to increase the number of observations (presumably to reduce the bias of statistical analysis and obtain more reliable conclusions).

Sunflower seeds, as food for the studied rodents, do not occur naturally in this region of the Alps. They were introduced as bait, so they are an external (additional) factor that could have influenced the obtained results. Increasing the availability and quantity of food only in the areas monitored by the cameras may have contributed to a higher than normal number of observed/camera-recorded rodents (especially mice) attracted to the readily available food. As a result, the number of encounters between mice and bank voles may have increased, which over a longer period of time (e.g., a two-year study) may have resulted in an inverse activity pattern in bank voles (or make a significant contribution to it). To exclude such a scenario, it was necessary to check whether the bait had a significant effect or not on the results obtained. For this purpose, preliminary studies could be carried out to test the significance of the effect of bait (sunflower seeds) on the nocturnal activity of the tested rodents (places monitored by cameras with bait vs. places monitored by cameras without bait), or add an additional independent variable (bait/no bait) to the multivariate logistic regression model (similar to mast year/non mast year, Table 1). In the second approach, some of the places monitored by the cameras should be left without bait. Doubts related to the use of bait should be discussed and dispelled before the article is accepted for publication (otherwise, the conclusions drawn may be questioned).

Reviewer 2 Report

You have solely used camera traps to infer competition between Apodemus and Clethrionomys. I found your study interesting, but incomplete. You have no idea on the size of their populations, their breeding situation (reproductive vs nonbreeding), their age structure, and their locations within the forest. Thus is provides an extremely limited window into the ecology of these 2 genera.

1. It would have been much better if you had simultaneously live-trapped, marked and quantified the 2 genera so that you had an independent assessment of who is there, how many, and where. As you indicate, the habitat is marginal for Cleths (Low cover where the cameras were located). In addition, you quantified masting from pollen, not seed fall.  You should have had seed traps to rigorously quantified seed fall.

In addition, the obvious experiment was to do a removal experiment.  Remove Apodemus to assess the impact on Cleths.  The natural experiment (high vs low seedfall years) helps a bit

2. you used sunflower seeds to entice both species to your camera locations. What rigorous evidence is there that both species eat these? Especially since Cleths to not eat seeds from the mast trees

3. Cleth hits are independent of seed fall or of Apodemus presence. I.e. Cleths hit rates do not change from good to poor seed fall years.  Hence, the latter is not dependent on the former

4. thus the key question that you do not address is what are they completing for and why??  Not food, but apparently time when active.  Time is not a substitute variable for food, nest sites, etc.  What is the purpose of Apodemus' aggression?.  Even in your study you see little evidence for it.  Andrzejewski has a bit of evidence for it, but not much.

5. What other environmental forces could drive this 24 h separation between the 2 genera? The most obvious one is predation.  True you have only 7 hits with predators at the sites you have your cameras, but that could be an artifact of where you placed and pointed your cameras. Or that many of the predators have been all been killed or removed by humans.  In addition, avian predators (hawks and owls), are likely major predators that your cameras may not see/pick up.  The colouration of Cleths protects them from above; Apodemus has no such advantage, and hence must forage at night.

6. So the key unknown is the object/purpose of competition / spacing behavior. 

7. Finally discuss where we go from here.  What should you have done at the onset to rigorously quantify the limitation of one species by another.

Reviewer 3 Report

Manuscript ID: animals-2230884Title: High Frequency of Apodemus spp. mice boosts Inverse Activity Patternof Bank Vole Clethrionomys glareolus by Non-aggressive Intraguild Competitionin an Inner-Alpine Mixed-ForestAuthors: Remo Probst *, Renate Probst

Review

I found this manuscript nearly perfect – neatly written and based on extremely good literature, not speaking about clear experimental approach and statistic. Authors analyse classic question – how can sympatric animals share ecological niche? They investigated intra-guild competition of the bank voles and two species of Apodemus mice. I could recommend accepting this manuscript as is, but there are some small remarks.

Title is very long. Can authors agree to delete “by Non-aggressive Intraguild Competition in an Inner-Alpine Mixed-Forest”, or at least “by in an Inner-Alpine Mixed-Forest”? This wouild make title more readable.

Line 14: “mice of the genus Apodemus” – this is much better, than “Apodemus ssp.” Used many time across the main text. Please consider put at Line 41 (further referred as Apodemus mice).

2.3 chapter: I propose do not use 2.3.1–2.3.9 subchapters, just the plain text with underlines, if needed, or the similar syntax.

Line 167: 0.25 cm or 0.25 m?

Lines 222, 264 and further: remove “Mann-Withney-U-Test:”, it was explained in methods.

Line 244: what is (1) after chi-square – degrees of freedom?

Line 281: what is (2) after chi-square – degrees of freedom?

Lines 345, 441, 449, 482: journal in italics, please

Line 619: mistype

Round 2

Reviewer 2 Report

Thank you for addressing most of my concerns.  There are marked limitations of your study as I pointed out and you have now done so.  The critical thing is to know and address what Apodemus spp and Cleths are competing for.  It is not food.  Space - but why?  What is the mechanism. The other thing you should have speculated on was nocturnal avian predators (owls) and how these could drive the diurnal behavior of the Cleths.